# Hi-Gaussian: Hierarchical Gaussians under Normalized Spherical Projection for Single-View 3D Reconstruction

## Abstract

Single-view 3D reconstruction is a fundamental problem in computer vision, having a significant impact on downstream tasks such as Autonomous Driving, Virtual Reality and Augment Reality. However, existing single-view reconstruction methods are unable to reconstruct the regions outside the input field-of-view or the areas occluded by visible parts. In this paper, we propose Hi-Gaussian, which employs feed-forward 3D Gaussians for efficient and generalizable single-view 3D reconstruction. A Normalized Spherical Projection module is introduced following an Encoder-Decoder network in our model, assigning a larger range to the transformed spherical coordinates, which can enlarge the field of view during scene reconstruction. Besides, to reconstruct occluded regions behind the visible part, we introduce a novel Hierarchical Gaussian Sampling strategy, utilizing two layers of Gaussians to hierarchically represent 3D scenes. We first use a pre-trained monocular depth estimation model to provide depth initialization for *leader* Gaussians, and then leverage the *leader* Gaussians to estimate the distribution followed by *follower* Gaussians. Extensive experiments show that our method outperforms other methods for scene reconstruction and novel view synthesis, on both outdoor and indoor datasets.

## 1 Introduction

Living in a complex 3D environment, humans need to comprehend their surroundings to make even the slightest movements. Physiological studies show that humans can perceive depth even with monocular vision (Koenderink et al., 1995). Although binocular vision is more mature currently, it tends to degrade into monocular vision when sensing distant objects in autonomous driving scenarios. Moreover, in Virtual Reality and Augment Reality tasks, there is an expectation to reconstruct a scene from one single image captured by devices such as smartphones, which also makes single-view 3D reconstruction critically crucial. Unfortunately, reconstructing the 3D structure from a 2D image is ill-posed due to the lack of explicit geometric cues such as epipolar geometry, which are only available with two or more views. Therefore, single-view 3D reconstruction is an important but challenging issue in computer vision.

Single-view 3D reconstruction attracts more attention recently, with the development of differentiable rendering or neural rendering. PixelNeRF (Yu et al., 2021) introduces Neural Radiance Field into single-view reconstruction. It encodes a 2D image to obtain a feature volume, and finally decodes the feature volume into a radiance field using an MLP decoder. MINE (Li et al., 2021) uses multi-plane images to represent scenes, composed of RGB-$\alpha$ images at continuous depths, which enhances the scene representation capabilities. VisionNeRF (Lin et al., 2023) incorporates Transformer blocks into its encoder to better extract global features, which are combined with local features to improve the quality of reconstruction. Splatter Image (Szymanowicz et al., 2024b) introduces Gaussian Splatting into single-view reconstruction, addressing the challenge of the above methods that require costly point sampling and querying during novel view synthesis. It utilizes a U-Net to encode the input image into feature maps which is similar to the previous works, and then decodes the feature maps into attributes of Gaussians instead of implicit radiance fields, obtaining a scene represented by 3D Gaussians.

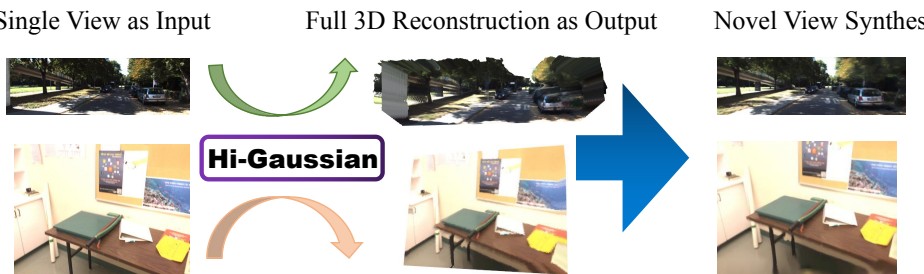

Figure 1: This is the pipeline of Hi-Gaussian, which requires only one single-view image as input, and reconstructs a 3D scene in a feed-forward manner. It can also render realistic images from novel views.

However, these methods only obtain features within the field-of-view (FOV) of the input view through their image encoders, without additional focus on regions outside the input view. This limits the ability of these methods to reconstruct areas beyond the input FOV, resulting in noticeable blurriness or holes when there is a significant difference between the rendering view and the input view. Another problem is their inability to reconstruct occluded regions. For example, as a typical 3DGS-based method, Splatter Image predicts only one Gaussian for each pixel, causing the Gaussians to cluster near visible surfaces, which limits the reconstruction of occluded regions. These drawbacks result in poor performances of the above methods when reconstructing complex scenes.

To address the aforementioned issues, we propose a simple and novel single-view reconstruction method named Hi-Gaussian. Hi-Gaussian requires only one image as input, and directly reconstructs a 3D scene in a feed-forward manner, as shown in Fig. 1. Our method is based on two key ideas. First, we design a Normalized Spherical Projection module, which is aimed at reconstructing scenes beyond the input FOV. After our Encoder-Decoder outputs feature maps, we project every 2D pixel to its normalized latitude-longitude spherical coordinates and uniformly resample the feature maps based on the spherical coordinates, which induces less distortion. Meanwhile, we expand the FOV of the feature maps by assigning a larger range of values to the spherical coordinates, allowing 3D Gaussians to cover a larger area. Second, we introduce a novel Hierarchical Gaussian Sampling strategy, which helps to reconstruct the regions occluded by the visible parts. Specifically, we estimate *leader* Gaussians through our pretrained depth network and Encoder-Decoder network, and then predict *follower* Gaussians based on the *leader* Gaussians. We assume that these *follower* Gaussians follow a normal distribution, whose parameters vary across *leader* Gaussians. We estimate the mean and covariance of the normal distribution and sample them from this distribution. Our main contributions are summarized as follows:

- We propose a Normalized Spherical Projection module to expand the input FOV, enabling reconstruction beyond the image FOV while inducing less projection distortion.

- We propose a Hierarchical Gaussian Sampling strategy to reconstruct occluded areas in images, enhancing reconstruction fidelity and improving performance on novel view synthesis.

- Experimental results show that the proposed method achieves state-of-the-art reconstruction accuracy on both outdoor and indoor datasets.

## 2 RELATED WORK

### 2.1 IMPLICIT 3D SCENE REPRESENTATION

One of the most representatives of implicit 3D scene representation is Neural Radiance Field (NeRF) (Mildenhall et al., 2021; Müller et al., 2022), which has seen widespread development in recent years due to its end-to-end differentiable nature and simple structure. NeRF utilizes a coordinate-based neural network to represent the scene, and typically start training from images with poses while employing differentiable volume rendering to back-propagate photometric loss. Many following works have made improvements to NeRF in various aspects. For exam-

ple, NeRF++ (Zhang et al., 2020), MipNeRF (Barron et al., 2021) and MipNeRF-360 (Barron et al., 2022) introduce advanced point sampling techniques, improving the rendering quality. PixelNeRF (Yu et al., 2021), MVSNeRF (Chen et al., 2021), SinNeRF (Xu et al., 2022) and SparseNeRF (Wang et al., 2023) enable NeRF to perform sparse-view 3D reconstruction while VisionNeRF (Lin et al., 2023) and SceneRF (Cao & de Charette, 2023) extend NeRF to single input view. However, due to implicit nature of Neural Radiance Field in storing scene information through MLPs, explicit scenes cannot be directly obtained. Additionally, the volume rendering process involves sampling and querying multiple points along rays, leading to a very slow rendering speed. Therefore, we do not choose implicit Neural Radiance Field as the scene representation but use an explicit representation instead.

## 2.2 EXPLICIT 3D SCENE REPRESENTATION

Explicit representation has been studied for decades due to its simplicity and intuitiveness. Many works explicit 3D scene representations including point clouds (Fan et al., 2017; Yang et al., 2019; Wiles et al., 2020; Vahdat et al., 2022), meshes (Kanazawa et al., 2018; Wang et al., 2018; Gao et al., 2022), and voxels (Choy et al., 2016; Girdhar et al., 2016; Yagubbayli et al., 2021). However, these representations are not smooth enough, which may affect the quality of novel view synthesis. The recently emerged method of 3D Gaussian Splatting (3DGS) (Kerbl et al., 2023) has become another mainstream approach for 3D reconstruction due to its continuous 3D representation and rapid differentiable rasterization. Among 3DGS-based methods, pixelSplat (Charatan et al., 2024), latentSplat (Wewer et al., 2024), MVSplat (Chen et al., 2024), and FreeSplat (Wang et al., 2024) extend 3DGS to sparse-view reconstruction. Whereas TriplaneGaussian (Zou et al., 2024), Splatter Image (Szymanowicz et al., 2024b), and AGG (Xu et al., 2024) further expand 3DGS to single-view object reconstruction. Our method also introduces 3D Gaussian Splatting into single-view reconstruction, primarily for scene reconstruction rather than object reconstruction, which is a much harder problem due to the complexity, diversity, and large scale of the scene.

## 2.3 SINGLE VIEW 3D RECONSTRUCTION AND NOVEL VIEW SYNTHESIS

Single-view 3D reconstruction involves feeding a single image into a neural network in a feed-forward manner to directly output a 3D scene. Among single-view 3D scene reconstruction methods, SynSin (Wiles et al., 2020) utilizes point clouds to represent scenes, requiring a GAN generator (Goodfellow et al., 2020) for image synthesis after projecting the point clouds onto the image plane, while MINE (Li et al., 2021) and MPI (Tucker & Snavely, 2020) use multi-plane images to represent scenes. In contrast, SRN (Sitzmann et al., 2019), pixelNeRF (Yu et al., 2021) and SceneRF (Cao & de Charette, 2023) use Neural Radiance Fields for scene representation. They both construct pixel-aligned feature maps using an image encoder, then get point-wise features by projecting 3D points onto the feature maps, and finally decode them using an MLP to obtain colors and opacities. These NeRF-based methods exhibit slow inference speeds due to volume rendering. Following the emergence of 3DGS (Kerbl et al., 2023), Splatter Image (Szymanowicz et al., 2024b) is designed to predict 3D Gaussians for reconstruction. However, Splatter Image estimates per-pixel aligned 3D Gaussians, limiting its capability to estimate only the surfaces of visible objects in the image, rather than unseen parts. To address this issue, we first design a Normalized Spherical Projection module to reconstruct the areas outside the input field-of-view, and then propose a Hierarchical Gaussian Sampling strategy to reconstruct the occluded regions.

## 3 THE PROPOSED HI-GAUSSIAN

Hi-Gaussian reconstructs the entire 3D scene from a single-view and is trained end-to-end only with posed images. Given a sequence of $K$ posed images $\mathbb{I} = \{(\mathbf{I}_1, \mathbf{P}_1), (\mathbf{I}_2, \mathbf{P}_2), \cdots, (\mathbf{I}_K, \mathbf{P}_K)\}$, we use $\mathbf{I}_1$ as input to construct 3D Gaussians $\{\mathbf{G}_i\}_{i=1}^N$ through a feed-forward network $f$. We denote it as $\{\mathbf{G}_i\}_{i=1}^N = f(\mathbf{I}_1)$. The remaining frames of the image collection (i.e., $\{\mathbf{I}_2, \cdots, \mathbf{I}_N\}$) serve as supervision for training losses. To begin with, we introduce the background of 3DGS for single-view reconstruction in Sec. 3.1. After that, we introduce our method and detail two key elements. Firstly, to extend the scene *beyond* input FOV, we introduce Spherical Projection in Sec. 3.2. Secondly, to extend the scene *behind* the visible part, we introduce the Hierarchical Gaussian Sampling in Sec. 3.3. Finally, we discuss our loss fuctions used during training in Sec. 3.4.

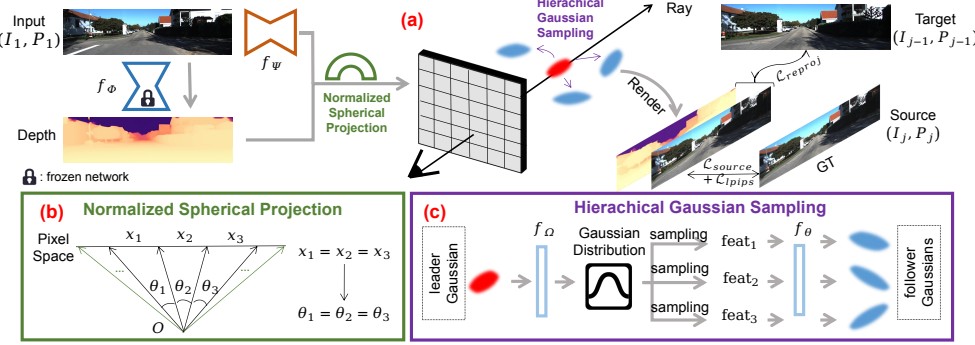

Figure 2: Overview of our Hi-Gaussian framework: (a) Given the first input image $\mathbf{I}_1$ from the image sequence $\mathbb{I}$, we first obtain the corresponding depth map using a pre-trained monocular depth estimation network $f_\Phi$. With known camera intrinsics, back-projection is utilized to derive 3D locations for each *leader* Gaussians. Meanwhile we input $\mathbf{I}_1$ into an encoder-decoder network $f_\Psi$ to predict *leader* Gaussians' other attributes. (b) To hallucinate colors and depth beyond input FOV, a Normalized Spherical Projection module is designed to extend 3D Gaussians to regions outside the input viewpoint. (c) To predict unseen parts behind the visible surfaces, we perform Hierarchical Gaussian Sampling based on *leader* Gaussians $\mathbf{G}^p$ to obtain *follower* Gaussians $\mathbf{G}^c$.

## 3.1 3DGS FOR SINGLE VIEW RECONSTRUCTION

**3D Gaussians as scene representations.** A set of 3D Gaussians $\{\mathbf{G}_i\}_{i=1}^N = \{(\boldsymbol{\mu}_i, \alpha_i, \mathbf{S}_i, \mathbf{R}_i, \mathbf{c}_i)\}_{i=1}^N$ represents a 3D scene (Kerbl et al., 2023). Here, $\boldsymbol{\mu}_i \in \mathbb{R}^3$ is the center position, $\alpha_i \in [0, 1]$ is the opacity, $\mathbf{S}_i \in \mathbb{R}^{3\times3}$ is Gaussian scale which is represented by a vector $\mathbf{s}_i \in \mathbb{R}^3$, $\mathbf{R}_i \in \mathbb{R}^{3\times3}$ is Gaussian orientation which can be obtained from a quaternion $\mathbf{q}_i \in \mathbb{R}^4$, and $\mathbf{c}_i \in \mathbb{R}^K$ is the spherical harmonics for view-dependent colors. These Gaussians can be rendered into images from arbitrary viewpoints. Let the un-normalized Gaussian function be defined as

$$g_i(\mathbf{x}) = \exp[-\frac{1}{2}(\mathbf{x} - \mu_i)^\top \boldsymbol{\Sigma}_i^{-1}(\mathbf{x} - \mu_i)] \tag{1}$$

where $\boldsymbol{\Sigma}_i = \mathbf{R}_i \mathbf{S}_i \mathbf{S}_i^\top \mathbf{R}_i^\top$. Then we can obtain a pixel's color by $\alpha$-blending:

$$\mathbf{c} = \sum_i \alpha_i g_i(\mathbf{x}_i) \mathbf{c}_i(\mathbf{v}) \tag{2}$$

where $\mathbf{v} \in \mathbb{R}^2$ is the direction of the pixel's ray emitted from the camera and $\mathbf{x}_i \in \mathbb{R}^3$ is intersection point of the pixel's ray with the $i_{\text{th}}$ Gaussian primitive. We can also obtain a depth map in a similar way, just replacing $\mathbf{c}_i$ with the depth value $d_i \in \mathbb{R}$ in Eqn. 2. We express the process as

$$d = \sum_i \alpha_i g_i(\mathbf{x}_i) d_i \tag{3}$$

where $d_i$ is the depth of the $i_{\text{th}}$ Gaussian. Eqn. 2 and Eqn. 3 are involved in forming our loss function, which will be introduced in Sec. 3.4.

**Feed-forward single-view reconstruction.** In the original 3DGS framework (Kerbl et al., 2023), the Gaussians' parameters need to be trained from scratch when reconstructing a new scene. In contrast, we directly infer a 3D Gaussian representation without per-scene training. Specifically, an Encoder-Decoder network is utilized with an image $\mathbf{I}_1 \in \mathbb{R}^{3\times H\times W}$ as input and a tensor $\mathbf{t} \in \mathbb{R}^{C\times H\times W}$ as output. The tensor $\mathbf{t}$ consists of all Gaussians' attributes, including the opacity $\alpha$, the depth $d \in \mathbb{R}_+$, the offset $\boldsymbol{\Delta} \in \mathbb{R}^{3\times3}$, the covariance $\boldsymbol{\Sigma} \in \mathbb{R}^{3\times3}$ represented by rotation $\mathbf{q} \in \mathbb{R}^4$ and scale $\mathbf{s} \in \mathbb{R}^3$, and the spherical coefficients $\mathbf{c} \in \mathbb{R}^{3(L+1)^2}$ where $L$ is the order of the spherical harmonics. Each pixel $\mathbf{u} = (u, v, 1)^\top$ corresponds to one Gaussian, whose center position can be calculated by $\boldsymbol{\mu} = \mathbf{K}^{-1}\mathbf{u}d + \boldsymbol{\Delta}$, where $\mathbf{K}^{-1} \in \mathbb{R}^{3\times3}$ is the intrinsic matrix of the camera. Given the relative pose $\mathbf{P}_{1\rightarrow j}$ between the *source* view $\mathbf{P}_j$ and the input view $\mathbf{P}_1$, along with the camera intrinsics $\mathbf{K}$, the *source* view image $\hat{\mathbf{I}}_j$ can be rendered by the fast differentiable renderer $\mathcal{R}$, which

is provided by 3DGS (Kerbl et al., 2023) and is based on Eqn. 2. For sake of generality, we write the process above as:

$$\hat{\mathbf{I}}_j = \mathcal{R}(\{\mathbf{G}_i\}_{i=1}^N, \mathbf{K}, \mathbf{P}_{1 \to j}) \tag{4}$$

The network is trained by minimizing the photometric loss $\mathcal{L}_{source} = \mathbb{E}_j \|\hat{\mathbf{I}}_j - \mathbf{I}_j\|$. For higher fidelity, we employ a pretrained depth model (Piccinelli et al., 2024) to directly predict Gaussians' 3D locations, inspired by Szymanowicz et al. (2024a). Moreover, we propose a novel Normalized Spherical Projection module to reconstruct the scenes beyond the input FOV, by projecting 2D pixels to spherical coordinates with a larger range of values. And we present a Hierarchical Gaussian Sampling strategy to reconstruct the scenes behind the invisible parts, by initially predicting the first layer of Gaussians (called *leader* Gaussians) and then predicting the second layer of Gaussians (called *follower* Gaussians) based on the first layer. We elaborate on Normalized Spherical Projection and Hierarchical Gaussian Sampling in Sec.3.2 and Sec.3.3 respectively.

### 3.2 NORMALIZED SPHERICAL PROJECTION

Since 3D Gaussians are predicted by feeding the network with only images, the scenes represented by 3D Gaussians tend to be confined within the FOV of the images. For reconstructing the scenes beyond the image FOV, it is a good choice to enlarge the FOV after projecting Gaussian primitives to spherical coordinates because spherical projection induces less distortion than its planar counterpart (Salomon, 2007; Cao & de Charette, 2023). However, spherical projection entails more spatial compression, which leads to information loss and inevitably undermines the quality of scene reconstruction, which is illustrated in Fig. 8. In order to reduce distortions while suppressing spatial compression, we adopt a novel projection approach, called *Normalized* Spherical Projection, after our Encoder-Decoder network $f_\Psi$. Given that a *leader* Gaussian primitive $\mathbf{G}_i^p$ output by $f_\Psi$ corresponds to pixel coordinates $[u, v]^\top$, with known camera intrinsics $\mathbf{K}$, its coordinates in the camera coordinate system are $[x, y]^\top \sim \mathbf{K}^{-1}[u, v]^\top$. We denote $x_{max}$ and $y_{max}$ as the maximum of $x$ and $y$, and $x_{min}$ and $y_{min}$ as the minimum of $x$ and $y$ respectively. Then we normalize the coordinates by

$$\begin{bmatrix} \hat{x} \\ \hat{y} \end{bmatrix} = \begin{bmatrix} (x - \bar{x})/\tilde{x} \\ (y - \bar{y})/\tilde{y} \end{bmatrix} \tag{5}$$

where $\bar{x} = (x_{max} + x_{min})/2, \bar{y} = (y_{max} + y_{min})/2$ and $\tilde{x} = (x_{max} - x_{min})/2, \tilde{y} = (y_{max} - y_{min})/2$. Then we map the normalized Cartesian coordinates to the corresponding spherical coordinates. The projection is shown as:

$$\begin{bmatrix} \theta \\ \phi \end{bmatrix} = \begin{bmatrix} \pi - \arctan(\hat{x}^{-1}) \\ \arccos(-\hat{y}/r) \end{bmatrix} \tag{6}$$

where $r = \sqrt{\hat{x}^2 + \hat{y}^2 + 1}$. We assign $[\theta, \phi]^\top$ a $0.25\times$ broader range of values, which leads to a larger FOV. After the projection above, $[\theta, \phi]^\top$ are discretized and sampled uniformly.

Our Normalized Spherical Projection module allows *leader* Gaussians to be distributed more uniformly within the FOV, as illustrated in Fig. 2 (b). In addition, the expanded FOV enables Gaussians to extend beyond the edges of the image.

### 3.3 HIERARCHICAL GAUSSIAN SAMPLING

When predicting *leader* Gaussians, we obtain their initial positions $\boldsymbol{\mu}^p$ through a pre-trained depth estimation network $f_\Phi$ and estimate other attributes $(\Delta\boldsymbol{\mu}^p, \alpha^p, \mathbf{S}^p, \mathbf{R}^p, \mathbf{c}^p)$ through an encoder-decoder network $f_\Psi$, where $\Delta\boldsymbol{\mu}^p$ denotes the position offset of each *leader* Gaussian. $\Delta\boldsymbol{\mu}^p$ is expected to serve a dual purpose: accurately predicting positions of *leader* Gaussians for the visible regions and potentially allowing some *leader* Gaussians to move into occluded parts. However, our experiments show that only predicting $\Delta\boldsymbol{\mu}^p$ is not enough for reconstruction of the occluded regions. When only one Gaussian is estimated for each pixel, these Gaussians tend to aggregate near the surface of the visible parts. This phenomenon ensures geometrical fidelity of the visible parts but neglects the invisible parts, leading to compromised rendering quality under significant viewpoint differences, as shown in Fig. 4 and 5.

To better reconstruct occluded regions within the scene, we employ a variational approach to additionally estimate $M$ *follower* Gaussians based on each *leader* Gaussian, which is illustrated in Fig.

| Methods | SemanticKITTI | | | BundleFusion | | |
|---|---|---|---|---|---|---|
| | PSNR↑ | SSIM↑ | LPIPS↓ | PSNR↑ | SSIM↑ | LPIPS↓ |
| PixelNeRF (Yu et al., 2021) | 15.80 | 0.466 | 0.489 | 20.51 | 0.822 | 0.351 |
| MINE (Li et al., 2021) | 16.03 | 0.496 | 0.448 | 20.60 | 0.763 | 0.377 |
| VisionNeRF (Lin et al., 2023) | 16.49 | 0.483 | 0.468 | 20.51 | 0.831 | 0.332 |
| SceneRF (Cao & de Charette, 2023) | 16.46 | 0.482 | 0.476 | 25.07 | 0.853 | 0.323 |
| Splatter Image (Szymanowicz et al., 2024b) | 15.83 | 0.457 | 0.395 | 24.13 | 0.817 | 0.205 |
| Ours | **16.78** | **0.518** | **0.358** | **25.50** | **0.877** | **0.179** |

Table 1: Quantitative results on SemanticKITTI and BundleFusion datasets. We outperform all other methods on PSNR, SSIM and LPIPS.

2 (c). Specifically, we sample a latent vector $\mathbf{feat}^c$ for each *follower* Gaussian, assuming that $\mathbf{feat}^c$ follows a multivariate Gaussian distribution:

$$\mathbf{feat}^c \sim \mathcal{N}\left(\mathbf{G}^p, f_\Omega(\mathbf{G}^p)\right) \tag{7}$$

where the attributes of $\mathbf{G}^p$ are considered as the **mean**. And $\mathbf{G}^p$ is fed into a shallow MLP $f_\Omega$ with a single hidden layer to predict the **covariance**. Subsequently, we input $\mathbf{feat}^c$ into another shallow MLP $f_\Theta$ to obtain the corresponding *follower* Gaussian:

$$\mathbf{G}^c = f_\Theta(\mathbf{feat}^c) \tag{8}$$

After sampling $M$ latent vectors for each *leader* Gaussian, we get $M$ *follower* Gaussians. The final 3D scene representation consists of *leader* Gaussians and *follower* Gaussians. We denote it as $\{\mathbf{G}_i\}_{i=1}^N = \{\mathbf{G}_i^p\}_{i=1}^{N^p} \cup \{\mathbf{G}_i^c\}_{i=1}^{M \times N^p}$, where $N = (M+1) \times N^p$.

### 3.4 LOSS FUNCTIONS

Due to the randomness of Hierarchical Gaussian Sampling, the accuracy of the rendered images and depths may be slightly influenced. In order to further improve the quality of reconstruction and novel view synthesis, we introduce a reprojection loss following (Cao & de Charette, 2023) as below:

$$\mathcal{L}_{reproj} = \frac{1}{H \times W} \sum_{v=1}^{H} \sum_{u=1}^{W} \|\hat{\mathbf{I}}_{source}(u,v) - \mathbf{I}_{target}\left(proj\left(\hat{D}_{source}(u,v)\right)\right)\|_1 \tag{9}$$

where $H$ and $W$ are the height and width of the image respectively, $\hat{\mathbf{I}}_{source}$ is the image rendered under the source view through Eqn. 2, $\hat{D}_{source}$ is the depth rendered under the source view through Eqn. 3, $\mathbf{I}_{target}$ is the ground-truth image under the target view, and $proj(\cdot)$ is the transformation of $(u,v)$ from the source view to the target view. Besides, we adopt the RGB loss $\mathcal{L}_{source}$ and the perceptual loss $\mathcal{L}_{lpips}$ from Szymanowicz et al. (2024b). Note that these three loss functions are calculated under the source view, not making use of the input view image. Therefore, we render 3D Gaussians into the input view as $\hat{\mathbf{I}}_{input}$, constructing an $l$-1 RGB loss between $\hat{\mathbf{I}}_{input}$ and $\mathbf{I}_{input}$, which is formulated as $\mathcal{L}_{input} = \|\hat{\mathbf{I}}_{input} - \mathbf{I}_{input}\|_1$. Our total loss is as below:

$$\mathcal{L}_{total} = \lambda_1 \mathcal{L}_{source} + \lambda_2 \mathcal{L}_{reproj} + \lambda_3 \mathcal{L}_{lpips} + \lambda_4 \mathcal{L}_{input} \tag{10}$$

Our model is trained by minimizing $\mathcal{L}_{total}$. It can be viewed as a self-supervised manner, because we only use posed 2D images as ground truth for supervision, without 3D supervision.

## 4 EXPERIMENTS

In this section, we describe our experimental setup, evaluate our approach on novel view synthesis, and conduct ablation studies to validate our design choices.

### 4.1 EXPERIMENTAL SETUP

We evaluate Hi-Gaussian primarily on novel view synthesis. Our experiments are conducted on two datasets: *outdoor* SemanticKITTI (Geiger et al., 2012; Behley et al., 2019) and *indoor* BundleFusion (Dai et al., 2017). SemanticKITTI has large driving scenes encompassing challenging natural

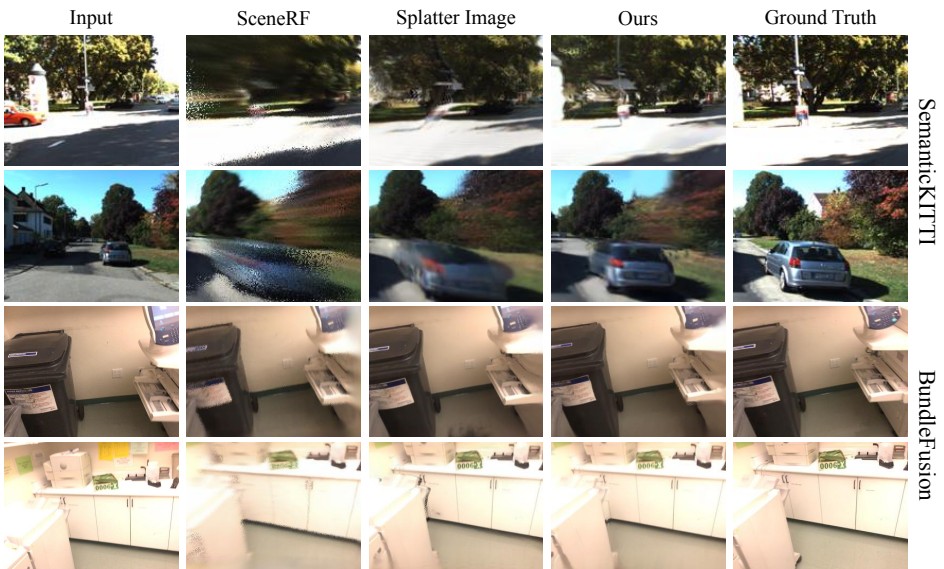

|  | Input | SceneRF | Splatter Image | Ours | Ground Truth |
|--|-------|---------|----------------|------|--------------|

Figure 3: Qualitative comparison on SemanticKITTI and BundleFusion. For aesthetic layout, we crop the results of SemanticKITTI to match the aspect ratio of BundleFusion. For both datasets, we note that our rendered images are sharper and more realistic than other methods.

environments such as trees, skies, along with dynamic elements like vehicles and pedestrians. Its image sequences are captured by a forward-facing camera with little rotation but large forward translation between viewpoints. In contrast, BundleFusion has shallow indoor scenes with pronounced lateral shifts between viewpoints. We follow the experimental settings from SceneRF (Cao & de Charette, 2023) for both SemanticKITTI and BundleFusion.

**Compared Methods.** We compare our method with five novel view synthesis approaches: pixelNeRF (Yu et al., 2021), MINE (Li et al., 2021), VisionNeRF (Lin et al., 2023), SceneRF (Cao & de Charette, 2023), which are NeRF-based methods, and Splatter Image (Szymanowicz et al., 2024b), which is a 3DGS-based method. For all compared methods, we use their official implementations.

**Evaluation Metrics.** To evaluate visual fidelity, we compare the rendered images of each method with their corresponding ground truth images. We use Peak Signal-to-Noise Ratio (PSNR, higher is better), Structural Similarity Index Measure (SSIM, higher is better) and the Learned Perceptual Image Patch Similarity (LPIPS, lower is better) as evaluation metrics.

**Implementation Details.** The pre-trained monocular depth estimation network $f_\phi$ in our model is from UniDepth (Piccinelli et al., 2024). Our Encoder-Decoder network is composed of a UNet (Song et al., 2020) and a $1 \times 1$ convolutional layer. In Normalized Spherical Projection module, we enlarge the input FOV by $0.25\times$. For Hierarchical Gaussian Sampling, $M$ is set to 3, which is adequate for scene representation. Our method is trained by minimizing the loss function of Eqn. 10 and set $\lambda_1 = \lambda_2 = \lambda_4 = 1$ and $\lambda_3 = 0.05$. We train our method and all compared methods using AdamW (Loshchilov, 2017) optimizer on 4 Tesla V100 32G with learning rate of 1e-5 for 50 epochs.

### 4.2 RESULTS

We evaluate the quality of reconstruction through novel view synthesis on SemanticKITTI and BundleFusion seperately. For SemanticKITTI, we input a single image each time and test every model with predicting the images within the next 10 meters. For BundleFusion, we input a single image each time and test every model with predicting the images from the 16 frames preceding it and the 16 frames following it. Quantitatively, we take the average of the evaluation results for each frame as the final metrics, as shown in Table 1. Our method achieves the best performance

| Methods | SemanticKITTI | | | BundleFusion | | |
|---|---|---|---|---|---|---|
| | PSNR↑ | SSIM↑ | LPIPS↓ | PSNR↑ | SSIM↑ | LPIPS↓ |
| w/o Normalized Spherical Projection | 16.47 | 0.497 | 0.364 | **25.69** | 0.875 | 0.187 |
| w/o Hierarchical Gaussian Sampling | 16.33 | 0.491 | 0.361 | 25.57 | 0.873 | 0.189 |
| Full settings | **16.78** | **0.518** | **0.358** | 25.50 | **0.877** | **0.179** |

Table 2: Architecture ablations on SemanticKITTI and BundleFusion. Both of the two key components contribute to better performance on Novel View Synthesis.

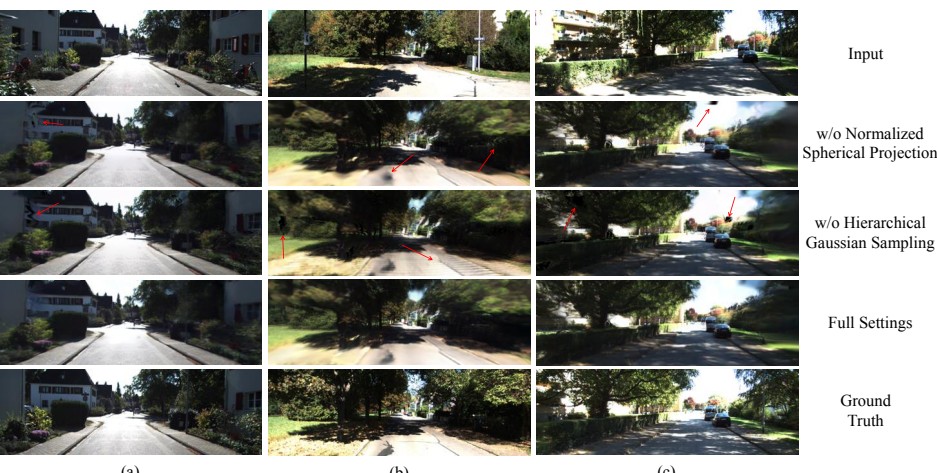

Figure 4: Ablations on SemanticKITTI (val). Both Normalized Spherical Projection module and Hierarchical Gaussian Sampling contribute to rendering novel view images with higher fidelity. Normalized Spherical Projection module helps to inpaint the holes beyond the input FOV (column c) and mitigate distortions (column a and b). Hierarchical Gaussian Sampling helps to reconstruct the occluded regions behind the trees (column b and c) and the building (column a).

across the three metrics on both the datasets. Qualitatively, we compare our method with a typical NeRF-based method SceneRF (Cao & de Charette, 2023) and a typical 3DGS-based method Splatter Image (Szymanowicz et al., 2024b). Fig. 3 displays the visual results. It is notable that when there is a significant difference between the rendering view and the input view, the images output by SceneRF exhibit noticeable blurriness, which is a common issue with NeRF methods due to limited point sampling in volume rendering. Additionally, SceneRF incorrectly renders numerous scattered points at object edges, leading to poor visual quality. Although Splatter Image outputs sharper and clearer images, it suffers from severe distortions due to the lack of depth constraints during training. Moreover, since Splatter Image predicts only one Gaussian per pixel, its Gaussians tend to cluster near visible surfaces. This makes itself unable to reconstruct occluded areas in the scene, especially in outdoor environments, resulting in holes and ripple artifacts in novel view images. On the contrary, our method renders images that are both sharp and realistic, with the best visual effects.

Furthermore, we compare the rendering quality of our method with other methods at various distances from the input view, as illustrated in Fig. 6. Although SceneRF performs best in terms of PSNR and SSIM when the distance between the novel view and the input view is close, our method quickly surpasses SceneRF and Splatter Image as the distance increases. This highlights the robustness of our approach to input view distances.

## 4.3 ABLATION STUDY

To further demonstrate the effectiveness of our method, we conduct ablation studies with or without certain components, *i.e.*, Normalized Spherical Projection and Hierarchical Gaussian Sampling, on the validation set of SemanticKITTI and BundleFusion. See quantitative results in Table 2 and qualitative ones in Fig. 4 and 5.

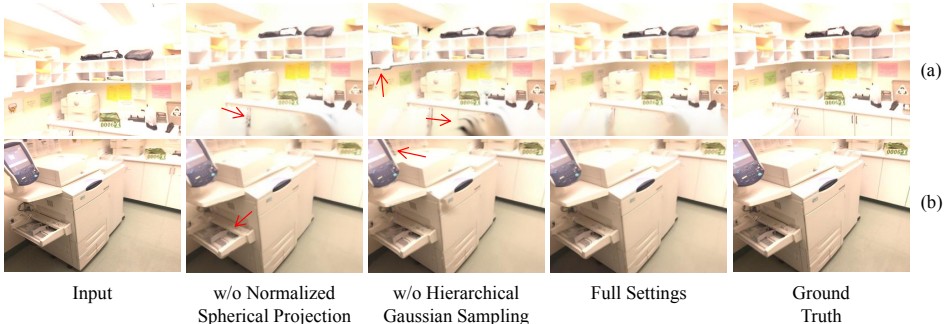

Figure 5: Ablations on BundleFusion (val). Without Normalized Spherical Projection, our model fails to reconstruct outside the input FOV, leading to ghosting artifacts. Without Hierarchical Gaussian Sampling, there will be ripple artifacts and holes in the rendered novel views.

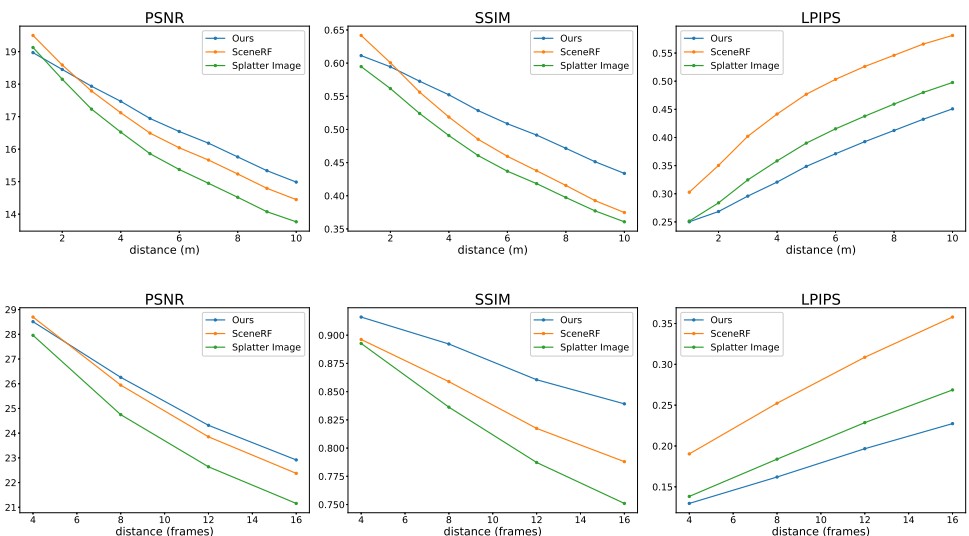

Figure 6: Performances of different methods at varying input view distances on SemanticKITTI (the first row) and BundleFusion (the second row). The quality of novel view synthesis drops as the distance increases due to lower overlaps of FOV with the input view.

**Importance of Normalized Spherical Projection.** To verify the effectiveness of our Normalized Spherical Projection module, we compare Hi-Gaussian to a variant (w/o Normalized Spherical Projection module) that eschews FOV enlargement with normalized spherical projection. Qualitatively, this produces holes (see Column c in Fig. 4) and ghosting artifacts (see Fig. 5) that are evidence of incorrect predictions of scenes. Quantitatively, this leads to a significant drop in performance. Moreover, in order to illustrate the effect of normalization, we compare our *normalized* spherical projection with the *un-normalized* one proposed by SceneRF (Cao & de Charette, 2023). For fairness, we fix other hyper-parameters, such as the FOV scaling factor and the number of Gaussians. We experiment with normalized/un-normalized spherical projection, getting respectively **16.78**/16.76 for PSNR, **0.518**/0.512 for SSIM and **0.358**/0.370 for LPIPS. The results demonstrate that our normalized spherical projection performs better than the un-normalized one. Fig. 8 also shows that normalized spherical projection induces less spatial compression than the un-normalized one, which slightly enhances the rendering quality.

**Importance of Hierarchical Gaussian Sampling.** To verify the effectiveness of our Hierarchical Gaussian Sampling, we remove the parts of our network that predict *follower* Gaussians. We use only one layer Gaussians (*i.e.*, *leader* Gaussians) for reconstruction. Qualitatively, Fig. 4 illustrates that ripple artifacts are significant when only predicting one layer Gaussians. Columns (a) and (c)

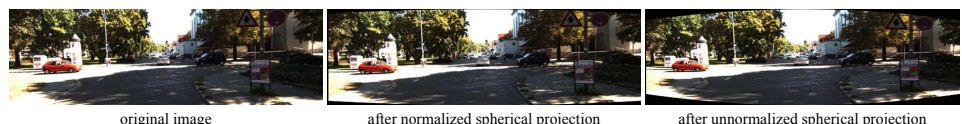

Figure 7: Visualized layers of Gaussians on SemanticKITTI (val). The red part denotes the *leader* Gaussians within the input FOV. The green part denotes the additional *leader* Gaussians from FOV expansion. And the blue part denotes the *follower* Gaussians. The additional *leader* Gaussians fill the regions beyond the input FOV. And the *follower* Gaussians fill occluded regions, such as the areas behind the skywalk or the trees.

Figure 8: For visualization, we apply two different spherical projections to an image. In practice, we perform spherical projections on high-dimensional features. The black areas around the images show that our *normalized* spherical projection induces less spatial compression than the *unnormalized* spherical projection.

respectively display the novel view synthesis of regions that are occluded by buildings/vehicles in the input view. Our model with full settings is capable of predicting and inpainting the occluded regions of the scene, while our model without Hierarchical Gaussian Sampling cannot predict them, resulting in significant holes. Fig. 5 also shows significant ripple artifacts and holes, due to obstruction by the cabinet/printer. Quantitatively, Table 2 indicates that the performance drops markedly without Hierarchical Gaussian Sampling.

**Analysis.** To investigate the role of Normalized Spherical Projection Module and Hierarchical Gaussian Sampling in reconstruction, we visualize layers of Gaussians on SemanticKITTI. Fig. 7 shows that *leader* Gaussians are primarily distributed on visible surfaces, while *follower* Gaussians are mainly distributed in occluded regions like the areas behind the skywalk or the trees. It is worth noting that when there is a significant rotation between the novel view and the input view, the FOV expansion in Normalized Spherical Projection Module aids in reconstructing areas beyond the input FOV.

## 5 CONCLUSION

In this paper, we introduce Hi-Gaussian, a feed-forward model that performs 3D reconstruction with high fidelity from a single image. The proposed Normalized Spherical Projection module projects 2D pixels to spherical coordinates with a larger range of values, aiding in the reconstruction of the scenes beyond the input FOV. And the proposed Hierarchical Gaussian Sampling strategy initially predicts *leader* Gaussians and then predicts *follower* Gaussians based on *leader* Gaussians, helping to reconstruct the occluded regions behind the visible parts. Both the two components enhance the quality of reconstruction and novel view synthesis. In two single-view 3D reconstruction tasks, which are outdoor SemanticKITTI and indoor BundleFusion respectively, our proposed approach achieves state-of-the-art performances.

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

# APPENDIX

## A   MORE EXPERIMENTS

Additional qualitative and quantitative results are presented to further elaborate on the superiority of our proposed method. We start by showcasing a more comprehensive comparison with other methods, then present a more detailed ablation study.

### A.1   QUALITATIVE COMPARISON WITH OTHER METHODS

**Cross-dataset Novel View Synthesis.**   To better demonstrate the generalization capability of our method, we conduct cross-dataset evaluations on novel view synthesis. Models are trained on BundleFusion and are tested on NeRF-LLFF dataset. The qualitative results in Fig. 9 indicate that our method renders the sharpest and clearest images in cross-dataset generalization.

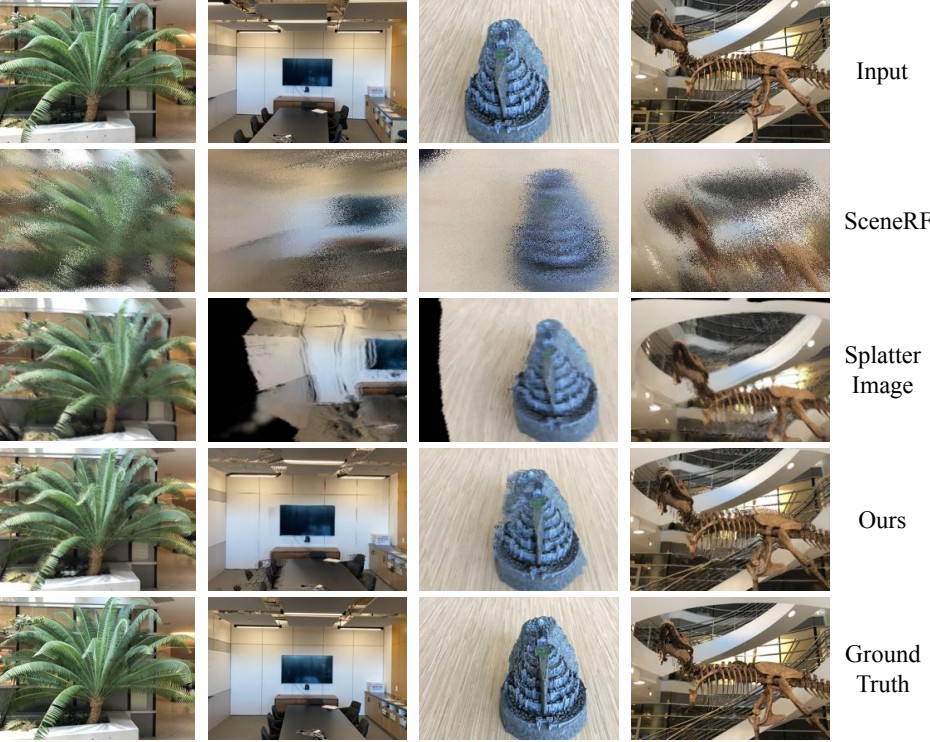

Figure 9: Qualitative evaluations on cross-dataset generalization from BundleFusion to NeRF-LLFF dataset.

**Mesh Visualization of Scene Reconstruction.**   To offer a more intuitive representation of scene reconstruction, we display 3D meshes on the validation set of SemanticKITTI and BundleFusion. These meshes are produced from the scene TSDF, which is obtained through the conversion of rendered images and depths. Fig. 10 and Fig. 12 demonstrate that our method reconstructs the clearest and sharpest meshes.

### A.2   QUANTITATIVE COMPARISON WITH OTHER METHODS

**Cross-dataset Novel View Synthesis.**   We present quantitative results on cross-dataset generalization. Models are trained on BundleFusion and are tested on all 8 sequences in NeRF-LLFF dataset. Table 3 shows that our method achieves the best performance on most sequences in NeRF-LLFF dataset.

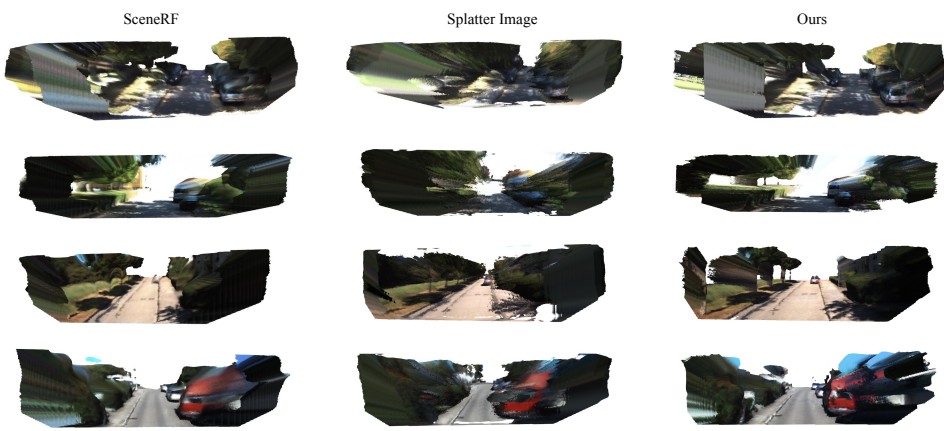

Figure 10: 3D meshes on SemanticKITTI (val.).

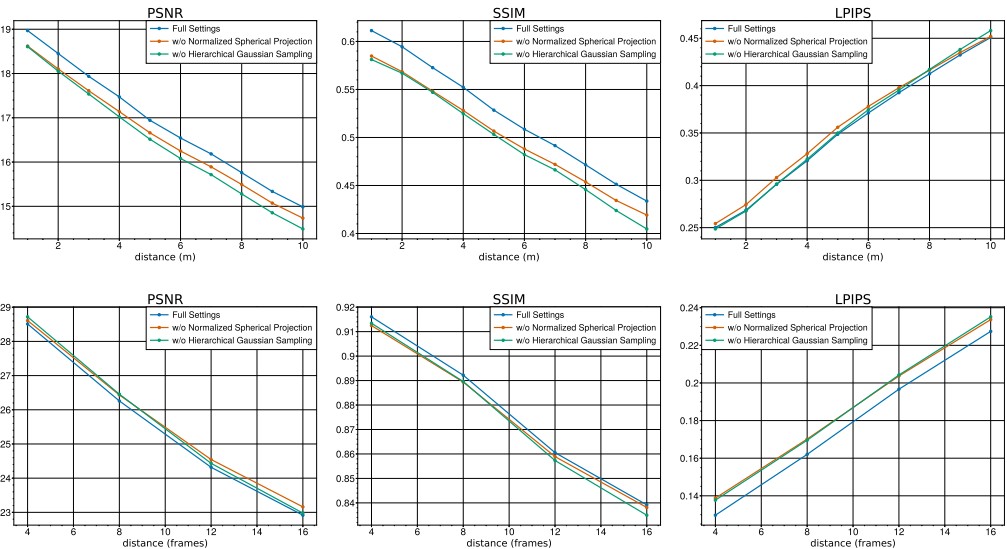

Figure 11: Performances at varying input view distances on SemanticKITTI (the first row) and BundleFusion (the second row) for ablation study. Our model with full settings usually achieves the best performance.

**Novel Depth Synthesis.** To better showcase the depth estimation capability of our method, we conduct evaluation on novel depth synthesis following the experimental setup of SceneRF (Cao & de Charette, 2023). Our approach outperforms other methods across all metrics, which is shown in Table 4 and Table 5.

**Scene Reconstruction.** We also evaluate reconstruction following SceneRF (Cao & de Charette, 2023) for a fair comparison. As demonstrated in Table 6, our method achieves the best reconstruction performance. It is worth noting that if we propose an improved scene reconstruction approach, we can get better results.

## A.3 ABLATION STUDY

To further demonstrate the effectiveness of our method, we conduct more ablation studies. We compare the quality of novel view synthesis at different distances from the input view. The results in Fig. 11 show that our Normalized Spherical Projection module and Hierarchical Gaussian Sampling

| Methods | **Fern** | | | **Flower** | | | **Fortress** | | | **Horns** | | |
|---|---|---|---|---|---|---|---|---|---|---|---|---|
| | PSNR↑ | SSIM↑ | LPIPS↓ | PSNR↑ | SSIM↑ | LPIPS↓ | PSNR↑ | SSIM↑ | LPIPS↓ | PSNR↑ | SSIM↑ | LPIPS↓ |
| SceneRF | 13.64 | 0.130 | 0.702 | 13.31 | **0.263** | 0.584 | 15.46 | 0.290 | 0.636 | 14.55 | 0.222 | 0.650 |
| Splatter Image | 11.89 | 0.133 | 0.610 | 12.36 | 0.107 | 0.523 | 8.95 | 0.218 | 0.565 | 14.48 | 0.266 | 0.469 |
| Ours | **16.93** | **0.439** | **0.359** | **14.01** | 0.167 | **0.468** | **17.59** | **0.302** | **0.318** | **15.84** | **0.398** | **0.379** |

| Methods | **Leaves** | | | **Orchids** | | | **Room** | | | **Trex** | | |
|---|---|---|---|---|---|---|---|---|---|---|---|---|
| | PSNR↑ | SSIM↑ | LPIPS↓ | PSNR↑ | SSIM↑ | LPIPS↓ | PSNR↑ | SSIM↑ | LPIPS↓ | PSNR↑ | SSIM↑ | LPIPS↓ |
| SceneRF | **13.77** | **0.161** | 0.548 | 11.31 | 0.128 | 0.716 | 12.74 | 0.286 | 0.743 | 10.65 | 0.093 | 0.757 |
| Splatter Image | 12.66 | 0.120 | 0.488 | 9.88 | 0.067 | 0.615 | 9.62 | 0.207 | 0.648 | 9.22 | 0.074 | 0.638 |
| Ours | 13.00 | 0.158 | **0.443** | **11.76** | **0.145** | **0.507** | **16.26** | **0.454** | **0.328** | **16.91** | **0.453** | **0.274** |

Table 3: Quantitative evaluations on cross-dataset generalization from BundleFusion to NeRF-LLFF dataset.

| Methods | **SemanticKITTI** | | | | | | |
|---|---|---|---|---|---|---|---|
| | Abs Rel↓ | Sq Rel↓ | RMSE↓ | RMSE log↓ | $\delta_1 \uparrow$ | $\delta_2 \uparrow$ | $\delta_3 \uparrow$ |
| PixelNeRF (Yu et al., 2021) | 0.2364 | 2.080 | 6.449 | 0.3354 | 65.81 | 85.43 | 92.90 |
| MINE (Li et al., 2021) | 0.2248 | 1.787 | 6.343 | 0.3283 | 65.87 | 85.52 | 93.30 |
| VisionNeRF (Lin et al., 2023) | 0.2054 | 1.490 | 5.841 | 0.3073 | 69.11 | 88.28 | 94.37 |
| SceneRF (Cao & de Charette, 2023) | 0.1681 | 1.291 | 5.781 | 0.2851 | 75.07 | 89.09 | 94.50 |
| SplatterImage (Szymanowicz et al., 2024b) | 0.2519 | 2.127 | 7.282 | 0.4205 | 58.41 | 79.30 | 89.02 |
| Ours | **0.1165** | **0.812** | **4.702** | **0.2397** | **80.99** | **90.02** | **94.67** |

Table 4: Novel depth synthesis on SemanticKITTI datasets.

strategy consistently enhance the performance of our model, regardless of the distance between the novel view and the input view.

# B  MORE THEORETICAL STUDY

## B.1  NORMALIZED SPHERICAL PROJECTION

In Sec. 3.2, we propose normalized spherical projection, which introduces less spatial compression than the un-normalized one. In other words, the curvature of the upper edges in the images obtained from normalized spherical projection is flatter, which is shown in Fig. 8. Now we provide a theoretical analysis of the phenomenon described above.

The formula for un-normalized spherical projection is given as follows:

$$\begin{bmatrix} \theta \\ \phi \end{bmatrix} = \begin{bmatrix} \pi - \arctan\left(x^{-1}\right) \\ \arccos\left(-y/r\right) \end{bmatrix} \qquad (11)$$

where $[x, y]^\top$ is coordinates in the camera coordinate system and $r = \sqrt{x^2 + y^2 + 1}$. We consider $\theta$ and $\phi$ as functions of $x$ and $y$, denoting them as $\theta = h_1(x, y)$ and $\phi = h_2(x, y)$ respectively. From Eqn. 11, we can see that $h_1(x, y) = \pi - \arctan\left(x^{-1}\right)$ is solely dependent on $x$, while $h_2(x, y) = \arccos\left(-y/r\right) = \arccos\left(-y/\sqrt{x^2 + y^2 + 1}\right)$ is dependent on both $x$ and $y$. Taking the partial derivative of $h_2(x, y)$ with respect to $x$, we obtain

$$\frac{\partial h_2(x, y)}{\partial x} = -\frac{xy}{(x^2 + y^2 + 1)\sqrt{x^2 + 1}} \begin{cases} > 0 & \text{if } xy < 0 \\ = 0 & \text{if } xy = 0 \\ < 0 & \text{if } xy > 0 \end{cases} \qquad (12)$$

Thus, for a given constant $y_0 > 0$, $h_2(x, y_0)$ is strictly monotonically increasing for $x \in (-\infty, 0)$, and strictly monotonically decreasing for $x \in (0, +\infty)$. Similarly, $h_2(x, -y_0)$ is strictly monotonically decreasing for $x \in (-\infty, 0)$, and strictly monotonically increasing for $x \in (0, +\infty)$. Moreover, since $h_2(x, y) = h_2(-x, y)$, it is evident that $h_2(x, y)$ is an even function with respect to $x$. Therefore, for any $x \in \mathbb{R}$, for any $\hat{x} \in \{x' \mid |x'| < |x|\}$, we have $h_2(\hat{x}, y_0) > h_2(\pm x, y_0)$ and $h_2(\hat{x}, -y_0) < h_2(\pm x, -y_0)$.

Since normalization typically transforms $x$ into a smaller $\hat{x}$, our normalized spherical projection maps Cartesian coordinates to a smaller range of spherical coordinates. This results in the upper and

| Methods | BundleFusion | | | | | | |
| --- | --- | --- | --- | --- | --- | --- | --- |
| | Abs Rel↓ | Sq Rel↓ | RMSE↓ | RMSE log↓ | $\delta_1 \uparrow$ | $\delta_2 \uparrow$ | $\delta_3 \uparrow$ |
| PixelNeRF (Yu et al., 2021) | 0.6029 | 2.312 | 1.750 | 0.5904 | 46.34 | 72.38 | 83.89 |
| MINE (Li et al., 2021) | 0.1839 | 0.098 | 0.386 | 0.2386 | 65.53 | 91.78 | 98.21 |
| VisionNeRF (Lin et al., 2023) | 0.5958 | 2.468 | 1.783 | 0.5586 | 55.47 | 79.29 | 86.68 |
| SceneRF (Cao & de Charette, 2023) | 0.1766 | 0.094 | 0.368 | 0.2100 | 72.71 | 94.89 | 99.23 |
| SplatterImage (Szymanowicz et al., 2024b) | 0.2407 | 0.142 | 0.454 | 0.2710 | 57.06 | 89.00 | 97.99 |
| Ours | 0.0792 | 0.041 | 0.225 | 0.1101 | 95.28 | 99.30 | 99.75 |

Table 5: Novel depth synthesis on BundleFusion datasets.

| Methods | SemanticKITTI | | | BundleFusion | | |
| --- | --- | --- | --- | --- | --- | --- |
| | IoU↑ | Prec.↑ | Rec.↑ | IoU↑ | Prec.↑ | Rec.↑ |
| SceneRF (Cao & de Charette, 2023) | 13.84 | 17.28 | 40.96 | 20.16 | 25.82 | 47.92 |
| Splatter Image (Szymanowicz et al., 2024b) | 10.30 | 11.30 | 53.93 | 13.89 | 22.22 | 27.04 |
| Ours | 15.56 | 17.39 | 59.72 | 40.42 | 48.91 | 69.96 |

Table 6: Reconstruction evaluations on SemanticKITTI and BundleFusion datasets. We outperform all other methods across all metrics.

lower edges of the images obtained from normalized spherical projection appear relatively flatter than the un-normalized one.

SceneRF            Splatter Image            Ours

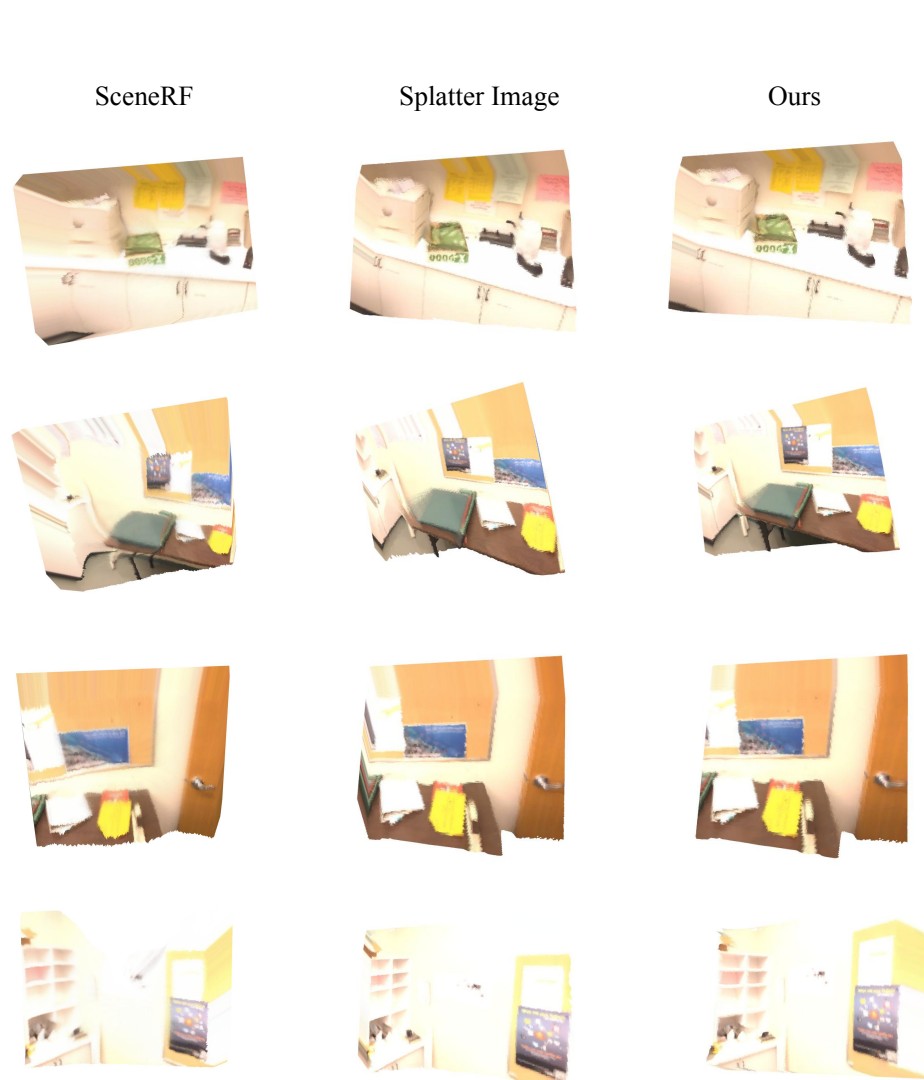

Figure 12: 3D meshes on BundleFusion (val.).

