# OpenReview forum: "Hi-Gaussian: Hierarchical Gaussians under Normalized Spherical Projection for Single-View 3D Reconstruction"
_ICLR.cc/2025/Conference — ICLR 2025 Conference Withdrawn Submission_

### Official Review · Reviewer_Sb4S · 2024-10-23

**Soundness:** 2
**Presentation:** 2
**Contribution:** 1
**Rating:** 5
**Confidence:** 4

**Summary:**

The paper studies the problem of single-view 3D reconstruction in computer vision. Specifically,  a pre-trained monocular depth estimation model is used to provide initializations for the Gaussians, and an Encoder-Decoder network is employed to predict 3D Gaussian attributes from single-image inputs. A Normalized Spherical Projection module is proposed to expand the input FOV and enable reconstruction beyond the FOV, also a Hierarchical Gaussian Sampling strategy is proposed to reconstruct the occluded areas in the image. Experiments show that the proposed method performs favorably against the other competitors on scene-level single-view reconstruction.

**Strengths:**

The presentation is in general clear and easy to follow. The investigated problem of single-view reconstruction is of general interest to a broad audience.

**Weaknesses:**

Although the paper has presented better experimental results against existing methods, some major concerns remain:

1. Among all the competitors, SceneRF is the most relevant, which is also dedicated to single-view reconstruction at scene level. However, by comparison, this work is much more restricted as it requires a pretrained depth estimation model while SceneRF is self-supervised. This  also renders the comparison between SceneRF and this work unfair in Table 1, due to the use of pretrained model.

2. The Normalized Spherical Projection module seems to be heavily influenced by the Spherical U-Net module in SceneRF, which again weakens the contribution of this paper.

3. The significance of single-view Gaussian Splatting does not entirely convince me. The occluded areas are unseen in a single view, and yet not hallucinated by any learned priors in the method as well. In a practical view, what is the significance of single-view Gaussian Splatting when occluded areas are visually blurry or even nonsense?

**Questions:**

Please respond to the three points made in the Weaknesses section.

---

> ### Comment · Reviewer_Sb4S · 2024-11-27
>
> Thanks for the detailed response. However, I still have some major concerns:
>
> A pre-trained depth estimation model is arguably better than using EfficientNet as an encoder in a scene reconstruction task. For this point, I am not convinced by authors' response. In addition, this method does not reconstruct occluded regions well due to its inherent limitation, as pointed out by reviewer Zfhn.
>
> In light of these concerns, I'll keep my original rating.

---

### Official Review · Reviewer_qLEv · 2024-10-26

**Soundness:** 4
**Presentation:** 4
**Contribution:** 3
**Rating:** 8
**Confidence:** 5

**Summary:**

The paper proposes a single-view 3D Gaussian Splatting (3DGS)-based 3D reconstruction method trained on a collection of posed images. This approach eliminates the need for expensive 3D ground-truth data for training and leverages the fast rendering capabilities of 3DGS. The method introduces two key contributions: a normalized spherical projection, which models information outside the field of view (FOV) while reducing the information compression typical of this projection, and hierarchical Gaussian sampling to effectively model occluded areas. The approach achieves SOTA performance in novel view synthesis in both outdoor and indoor datasets.

**Strengths:**

- The paper is well-written and clearly presented.
- The technical contributions are solid: Hierarchical Gaussian Sampling effectively models occluded areas—a critical challenge in single-view 3D reconstruction -- while the normalized spherical projection captures outside-FOV information, addressing limitations of the spherical projection used in existing techniques. Note that this Gaussian refers to the one used in 3DGS for rendering, not the Gaussian in SceneRF, which is only used to improve point sampling for NeRF.
- The experiments, including both the main results and ablation studies on novel view synthesis, are comprehensive and demonstrate the advantages of the proposed approach.
- Figures 8 and 9 are particularly helpful to understand the impact of the proposed components.

**Weaknesses:**

- The paper claims to address 3D reconstruction from a single image; however, it lacks any evaluation of 3D reconstruction. The paper should incorporate evaluations of 3D reconstruction and novel depth synthesis, similar to approaches like SceneRF, or include qualitative visualizations of 3D Gaussian distributions alongside depth maps, as demonstrated in pixelSplat, CVPR'24.

- The normalized spherical projection appears similar to the spherical projection technique proposed in SceneRF, with the primary difference being the normalization of x and y before projection. However, it's unclear how this normalization reduces spatial compression, as the paper suggests. What is the intuition behind this normalization?

- Figure 8 should be placed or at least referenced in Section 3.2 to illustrate the effect of normalization in spherical projection. Adding an explanation of the observed impact, specifically how normalization reduces black areas in the image, would help the readers understand more about the effect of the normalized spherical projection.

- Regarding Hierarchical Gaussian Sampling, since "leader" Gaussians are initialized using depth estimation, they are positioned on visible surfaces, with their distribution predominantly around these areas. It's unclear how "follower" Gaussians diverge from the "leader" Gaussian to cover occluded regions, given that they are sampled using the leader's distribution.

- Figure 7 would benefit from an additional column to highlight occluded scenery and areas outside the field of view in the novel view.

- The authors should clarify that the paper addresses single-view 3D reconstruction within a self-supervised setting, distinguishing it from monocular 3D reconstruction methods that often require 3D ground truth for supervision. I also think this is another selling point of the method and should be highlighted.

- On line 116, the authors mention choosing 3DGS over NeRF to achieve faster rendering speeds. Demonstrating the method's rendering speed relative to NeRF-based counterparts would provide valuable reference.

**Questions:**

Please refer to the Weaknesses section. I combined Weaknesses and Questions since they are linked and need to be together for better clarity.

---

### Official Review · Reviewer_Zfhn · 2024-10-28

**Soundness:** 2
**Presentation:** 2
**Contribution:** 2
**Rating:** 5
**Confidence:** 4

**Summary:**

They propose a generalizable single-view 3D reconstruction method with Normalized Spherical Projection module for larger FOV reconstrcution. Some validation results show the effectiveness.

**Strengths:**

The paper proposes to use normalized spherical projection for single view acne reconstruction and have some better results compared with baseline methods even though with blurry effects.

**Weaknesses:**

1.The approach to synthesizing unseen regions would benefit from a comparison with recent methods that utilize generative priors, particularly video diffusion models. Works like ViewCrafter and MotionCtrl demonstrate the effectiveness of such priors for single-view scene synthesis (Fig. 3 in ViewCrafter paper https://arxiv.org/pdf/2409.02048). Could the authors discuss how their method compares to these approaches, both theoretically and in terms of results?

2.The novel view demonstrations in Figure 3 primarily show viewpoint changes with substantial overlap with the input views (zoomed perspectives in the first two examples, and minimal FOV changes in the third and fourth). To better evaluate the method's capability in handling unseen regions, please consider including examples with more dramatic viewpoint changes."

3.The experimental comparison would be strengthened by including recent diffusion-based methods such as LucidDreamer, ViewCrafter, and MotionCtrl. This would help position the work within the current state-of-the-art in novel view synthesis.

4.The visual quality in Figure 3 shows some blurriness. Additionally, including rendered video results with camera trajectories would help better demonstrate the temporal consistency and overall quality of the synthesized views.

5.The robustness of the method could be better demonstrated through cross-dataset evaluation. This would help readers understand the generalization capabilities of the approach across different types of scenes and capture conditions.

**Questions:**

1.Do you retrain the compared methods with your evalution cameras for a fair comparison?

2.How you select the training and testing cameras?

---

### Official Review · Reviewer_ykBR · 2024-11-01

**Soundness:** 3
**Presentation:** 2
**Contribution:** 2
**Rating:** 5
**Confidence:** 4

**Summary:**

The paper aims at the single-view 3D reconstruction which follows the Splatter Image [Stanislaw et al] and proposes two techniques to solve the limited FOV and occlusions. To give a broader view range, the paper converts the pixel coordinates to a normalized  Spherical Projection and then enlarges the FOV during the projection. For occlusion, the paper splats more gaussians biased from the leader gaussians that come from the per-pixel splat. The experiments show good performances in outdoor datasets.

**Strengths:**

1. Quantitatively, the paper outperforms all other methods in semantic KITTI dataset and BundleFusion dataset.
2. The method gives a clear rendering in unbounded street views.

**Weaknesses:**

1. The differences between the paper and the splatter image are using a pre-trained monocular depth network, changing the pixel coordinate representation as spherical coordinate and splatting more gaussians using a leader and follower way. These contributions are a little incremental and do not show good insights into this field.
2. I argue the performances may highly depend on the pre-trained monocular depth. Please show more experiments that the paper’s ability to tolerate the bad depth prediction in some cases where UniDepth does not work. Or show more results for the zero-show generalization ability on other images like tank and temple, LLFF dataset.
3.  The qualitative results are not enough. For example, only the zoom-in effect is shown and the view range in indoor room is also limited. Moreover, no videos are present to see the 3D consistency of reconstruction.

**Questions:**

I mainly care about how much the paper depends on monocular depth estimation and generalization ability. Moreover, more qualitative results with arbitrary camera trajectory are also expected

---

> ### Comment · Reviewer_ykBR · 2024-11-25
>
> Thanks for the author's efforts!
>
> The rebuttal solves my concerns. However, I am still worried about the limited view range of the paper's NVS results and no video shows the paper's comparisons.
>
> Besides, the quantitative comparison in LLFF datasets is quite low. I wonder if the 3DGS plus UniDepth can reach better results in single-view reconstruction on LLFF dataset.

---

> ### Comment · Reviewer_qLEv · 2024-11-25
> **My opinion on Zero-Shot Performance Concerns in LLFF Dataset Evaluation**
>
> Dear Reviewer ykBR,
>
> I would like to also comment on the concern regarding zero-shot performance on the LLFF dataset. While understanding a method's generalization capabilities can bring valuable insight, I believe using zero-shot performance as a primary rejection criterion may not be appropriate in this case, as zero-shot prediction was not among the paper's stated claims.
>
> Furthermore, the performance gap can be attributed to the small scale and the domain gap of the datasets:
> - BundleFusion contains only 7 indoor scenes
> - LLFF represents a significantly different domain
> - The limited training data scope and domain discrepancy makes it challenging for any method, including 3DGS + UniDepth, to effectively bridge this domain gap
>
> While zero-shot performance can provide useful insights into a method's robustness, I believe it is beyond the scope of the paper.

---

### Comment · Reviewer_qLEv · 2024-12-03
**Reflections on the Authors' Responses and Reviewer Discussions**

Dear Fellow Reviewers and ACs,

After carefully reviewing the revised paper, the reviews, the authors’ responses, and the feedback from other reviewers, here are my thoughts:

One argument raised against the paper is that it does not reconstruct occluded regions well (reviewers Zfhn and Sb4S). I would like to point out that, in the context of 3D reconstruction from a single input image, there are three relevant regions:
- The empty space between the camera and the visible surface.
- The visible space.
- The occluded regions.

In my opinion, the last region is less critical in the task of 3D reconstruction (compared to scene completion), as the primary focus should be on reconstructing the visible aspects of the scene. Evaluating the quality of 3D reconstruction methods requires assessing the overall quality of geometry in all these regions, as demonstrated quantitatively in the authors' rebuttal. Consequently, it would be unfair to reject the paper based solely on the reconstruction quality of occluded regions. Furthermore, the assertion that "the quality is not good enough" (i.e., not good enough in comparison to what?) is unreasonable, especially since the paper has shown superior results compared to all the baselines.

Another argument is that the paper is an incremental improvement over SceneRF (reviewer Sb4S), I am not really agree with this as it has several key differences:
- It is a 3D Gaussian Splatting (3DGS) method instead of a NeRF-based approach as in SceneRF.
- The sampling method proposed in this paper is fundamentally different from that in SceneRF. In SceneRF, Gaussian sampling is used solely to sample better points along rays. However, in this work, the Gaussian is an actual geometry element within the scene that contributes to rendering.
- The normalized spherical coordinate system, while a relatively simple improvement upon the spherical UNet in SceneRF, is still effective.

I agree with reviewer Zfhn that incorporating a diffusion prior might help improve the reconstruction of occluded regions. However, I believe this is not mandatory, as the model can learn this prior from the data. Additionally, a diffusion prior is more relevant for scene completion tasks rather than for 3D reconstruction.

I somewhat agree with the concern raised about using a "pre-trained depth estimation model," (reviewer Sb4S) as it might make the comparison somewhat unfair. However, as long as this model was not trained on the same dataset, I think it is still acceptable to leverage existing pre-trained methods to provide a better prior -- this is somehow similar to the use of a diffusion prior as reviewer Zfhn suggested. Whether this approach is fair remains debatable, and I leave it to the ACs to make the final judgment.


That said, given the scores from other reviewers, I believe the paper has a low chance of acceptance. However, this discussion has significantly improved the paper, and I hope the authors incorporate all the feedback provided. I sincerely wish the paper will pass in a future submission.

---

> ### Author Response · Authors · 2024-12-03
>
> Dear Reviewer qLEv,
>
> We sincerely appreciate your understanding and support!

---

### Note · Authors · 2025-03-05

I have read and agree with the venue's withdrawal policy on behalf of myself and my co-authors.

---

### Meta-Review · Area_Chair_gypt · 2024-12-17

**Metareview:**

This paper receives ratings of 5,5,8,5. The AC follows the recommendations of the reviewers to reject the paper. The main concerns on the paper are: 1) The contributions are a little incremental and do not show good insights into this field. 2) Performances may highly depend on the pre-trained monocular depth. Especially when compared to SceneRF which is self-supervised. 3) The Normalized Spherical Projection module seems to be heavily influenced by the Spherical U-Net module in SceneRF. 4) Experimental results are not convincing, e.g. cross-dataset evaluation not shown, only the zoom-in effect is shown, the view range in indoor room is also limited, etc.  5) The significance of single-view Gaussian Splatting is not convincing.

Although a reviewer is supportive of the paper, the strengths mentioned by this reviewer is insufficient to explain away the weaknesses brought up by the other reviewers. The rebuttal and discussion phases have not convinced the reviewers to raise the scores to positive.

**Additional Comments On Reviewer Discussion:**

The rebuttal and discussion phases have not convinced the reviewers to raise the scores to positive.

---

### Decision · Program_Chairs · 2025-01-22

Reject